# Extensive Genome Exploration of *Clostridium botulinum* Group III Field Strains

**DOI:** 10.3390/microorganisms9112347

**Published:** 2021-11-13

**Authors:** Silvia Fillo, Francesco Giordani, Elena Tonon, Ilenia Drigo, Anna Anselmo, Antonella Fortunato, Florigio Lista, Luca Bano

**Affiliations:** 1Army Medical Center, Scientific Department, 00184 Rome, Italy; franc.giordani@gmail.com (F.G.); annanselm@gmail.com (A.A.); antonellafortunato75@gmail.com (A.F.); romano.lista@gmail.com (F.L.); 2Diagnostic and Microbiology Laboratory, Istituto Zooprofilattico Sperimentale delle Venezie, 31020 Villorba di Treviso, Italy; etonon@izsvenezie.it (E.T.); idrigo@izsvenezie.it (I.D.); lbano@izsvenezie.it (L.B.)

**Keywords:** animal botulism, *Clostridium botulinum* group III, genotyping, epidemiology

## Abstract

In animals, botulism is commonly sustained by botulinum neurotoxin C, D or their mosaic variants, which are produced by anaerobic bacteria included in *Clostridium botulinum* group III. In this study, a WGS has been applied to a large collection of *C. botulinum* group III field strains in order to expand the knowledge on these BoNT-producing Clostridia and to evaluate the potentiality of this method for epidemiological investigations. Sixty field strains were submitted to WGS, and the results were analyzed with respect to epidemiological information and compared to published sequences. The strains were isolated from biological or environmental samples collected in animal botulism outbreaks which occurred in Italy from 2007 to 2016. The new sequenced strains belonged to subspecific groups, some of which were already defined, while others were newly characterized, peculiar to Italian strains and contained genomic features not yet observed. This included, in particular, two new flicC types (VI and VII) and new plasmids which widen the known plasmidome of the species. The extensive genome exploration shown in this study improves the *C. botulinum* and related species classification scheme, enriching it with new strains of rare genotypes and permitting the highest grade of discrimination among strains for forensic and epidemiological applications.

## 1. Introduction

The genus *Clostridium*, recognized to be a polyphyletic taxon [1,2], includes obligate anaerobic, Gram-positive, spore-forming, rod-shaped bacteria that inhabit soil, sewage, and marine sediments, the intestinal tracts of various animals and decaying animal and plant products. They are able to produce illnesses mediated by various toxins [3,4,5]. One of these toxins is the botulinum neurotoxin (BoNT), which causes botulism, a neuro-paralytic disease affecting both humans and animals [6,7,8]. Due to its potency as a poison, BoNT is classified as category A agent and is considered a potential biological weapon and consequently also a threat for its possible use in bioterrorism attacks [6].

Seven different BoNT serotypes (A-G) and one chimeric form (H) have been characterized by the continuous isolation of intratypic variants known as subtypes, indicated with an Arabic number (BoNT/A1, BoNT/A2, BoNT/B1, BoNT/B2, etc) [8].

The bacteria that produce BoNT are divided into six phylogenetic and phenotypic groups (named with roman numbers from I to VI) that do not collectively constitute a monophyletic unit and are sufficiently different to be considered distinct species [9,10,11,12]. Groups I-IV are designated with the binomial *Clostridium botulinum* (for group IV the designation *Clostridium argentinense* is also used); group V includes *Clostridium baratii* strains that produce BoNT type F and group VI includes only *Clostridium butyricum* strains that produce BoNT type E [13]. Recently, a new species-based nomenclature has been proposed [2].

Based on 16S rDNA, whole genome sequencing (WGS) studies, and MALDI TOF MS analysis, *C. botulinum* group III was suggested to be closely related to *Clostridium novyi* and *Clostridium haemolyticum* and tentatively included in the same novel group designated as *Clostridium novyi sensu lato* [14,15,16].

*C. botulinum* group III produces BoNT type C and D or, most often, their mosaic variants C/D or D/C. Mosaic BoNT D/C (BoNT/DC) is composed of two-thirds BoNT/D, including the light chain and the amino-terminal (HN) fragment of the heavy chain, and one-third BoNT/C, corresponding to the carboxyl-terminal (HC) portion of the heavy chain. Mosaic BoNT C/D (BoNT/CD) shares the light chain and the HN of the heavy chain with BoNT/C and the HC fragment with BoNT/D [17]. Mosaic subtypes also showed different toxic capabilities when compared with non-mosaic subtypes. Indeed, BoNT/CD were shown to be more lethal to chicken than the non-chimeric type C, whereas BoNT/DC appeared to be more toxic in mice than BoNT/D [18,19].

Even though BoNTs produced by *C. botulinum* group III have been suspected as the cause of human botulism, the presence of BoNTs type C, D, C/D and D/C has never been clearly demonstrated in biological samples of human origin [20,21,22]. On the contrary, BoNTs produced by *C. botulinum* group III are the most frequent serotypes responsible for botulism outbreaks occurring both in wild and farmed animal species. Worldwide, BoNT/CD is responsible for the majority of avian botulism outbreaks, with catastrophic losses in wild birds and, sporadically, in poultry [23,24,25]. BoNT/DC seems to be the most frequent serotype detected in bovine botulism episodes and, sporadically, in avian botulism [15,19,25]. After the discovery of the BoNT mosaic subtypes produced by *C. botulinum* group III, non-mosaic BoNT type C appeared to be involved only in sporadic outbreaks which occurred in cattle and never in birds [14,26,27].

Animal botulism represents a serious environmental and economic concern, since the outbreaks are often characterized by massive mortality, implying relevant economic losses when bovines or poultry are involved [28,29,30,31,32,33,34,35,36]. Moreover, the involvement of food-producing animals poses serious questions about possible public health risks for humans.

Given the considerable economic impact potentially provoked by animal botulism, BoNTs must also be regarded as biological weapons for terroristic attacks toward animals for food production that are, from a certain point of view, more difficult to deal with than terrorism directed toward humans, with greater economic consequences and less clues to distinguish a terrorism episode from natural outbreaks [37,38,39].

For its clinical veterinary relevance, *C. botulinum* group III has been the object of several studies aiming to analyze the sub-species’ genetic diversity and to develop protocols for epidemiological investigation by using different genotyping techniques, such as amplified fragment length polymorphism (AFLP) [40], pulsed field gel electrophoresis (PFGE) [25], random amplified polymorphic DNA (RAPD) [41], and more recently WGS [16,42,43], applied to strains isolated worldwide, mostly in Europe (Sweden, France and Spain). What globally emerged from these studies is that *C. novyi sensu lato* is clearly divided into four main lineages (named I-IV, with lineage I furtherly divided in IA and IB); lineage IA and IB are composed only of *C. botulinum* strains, while other lineages also include *C. novyi* (II, III and IV), and *C. haemolyticum* (II) [16]. Serotype is partially associated with group: the members of lineage IA prevalently produce BoNT/CD, while those in lineage IB prevalently produce D/C; strains producing BoNT/C are found in lineage II; and strains producing BoNT/D belong to lineages IA, IB and II [16,43]. The strains isolated in Europe from avian botulism belong mainly to lineage IA, and most of them form a cluster of very similar genomes originating from Sweden, France and Spain [25,41,43].

WGS has also allowed a genomic content and organization analysis of *C. novyi sensu lato* that has led to the discovery of a complex pan-plasmidome, constituted of a large number of plasmids or circular prophages, that are considered the major factor for horizontal gene transfer in this taxon [15,16,42]. It has also been demonstrated that when *C. botulinum* type C is cured of its prophage, it simultaneously ceases to produce toxins and this nontoxigenic culture can then be converted to another toxigenic bacterial species (e.g., *Clostridium novyi* type A or to toxigenic *C. botulinum* types C or D) [44]. For this reason it has been proposed to use the differences in plasmid content for sample genotyping [43].

Based on length and sequence variability of flagellin gene (*fliC*), previous studies have demonstrated that this locus can be used as a molecular target for epidemiological studies [13,45,46]. Until now, five different variants of the *fliC* sequence were described and named from fliC-I to -V in *C. botulinum* group III. This gene can be present as a single or multiple copies and is harboured on the chromosome. Moreover, concordance was described between the *fliC* and BoNT serotype; until now, *fliC*-I (816 bps) was found in all *C. botulinum* producing BoNT/CD, *fliC*-II (816 bps) was related to type C and strain D 1873, *fliC*-III (819 bps) was detected in the C/D Eklund strain, *fliC*-IV (1239 bps) was associated only with D/C strains and *fliC*-V (816 bps) was found only in type D strains (D-4947) [13].

The aim of our study was to expand the knowledge of the phylogenetic relationships among sub-specific groups and on the genomic organization of *C. botulinum* group III, to derive epidemiological insights from the genetic comparison of strains, and to identify specific genomic elements that could allow the development of diagnostic tools to be employed in deliberate or accidental botulism outbreaks. Comparative genomic analysis was performed, including 60 newly sequenced Italian strains together with 48 previously published genomes, focusing the attention on the phylogenetic relations among the genomes and on the characterization of the differences in the BoNT-gene-carrying plasmid in the *BoNT* gene, in the *fliC* gene and in mobile elements.

## 2. Materials and Methods

### 2.1. Strains

Sixty *C. botulinum* group III field strains were sequenced for the present study. The strains were isolated from biological or environmental samples collected in 32 outbreaks of animal botulism which occurred in Italy from 2007 and 2016. The majority of the strains (52/60) were isolated in outbreaks which occurred in regions of the northeast of Italy (Veneto, Friuli Venezia Giulia, Trentino Alto Adige and Emilia Romagna); 3 occurred in regions of the northwest (Lombardia), while 4 occurred in the center and 1 in the south of Italy (Puglia) (Table 1). Most of the strains were isolated from the northeast of Italy because this is the area of territorial competence of the Istituto Zooprofilattico Sperimentale delle Venezie (IZSVE). In 17 outbreaks, more than 1 strain was available due to being isolated from different subjects or from both animal and environmental samples (Table 1).

The diagnosis of botulism was based on the characteristic clinical symptoms, the presence of BoNT-encoding genes in biological samples and, in a restricted number of birds, on the detection of BoNTs in the sera of symptomatic animals. Strain isolation was performed in the Veterinary Diagnostic Laboratory of Treviso, IZSVE. To isolate the strains, a broth enrichment was performed in fortified cooked meat medium (FCMM) and, after appropriate incubation conditions, the FCMM broths were tested by PCR protocols for BoNT-encoding gene detection. PCR-positive broths were subsequently plated in egg yolk medium (EYA) and blood agar base (BAB), modified as described elsewhere [27]. The neurotoxin gene profile of the isolates was investigated using published PCR protocols, and the actual BoNT production was assessed by mouse bioassay, conducted on the filtered broth supernatants [18]. Pure toxigenic strains were then stored in Microbank at −80 °C until whole genome sequencing was performed.

Moreover, 48 previously published genomes belonging to *C. novyi sensu lato* (36 *C. botulinum*, 9 *C. novyi* and 3 C. *haemolyticum*) were included in the study (Appendix A).

### 2.2. Growth Condition, DNA Isolation and WGS

The strains sequenced for this study were cultured in Cardella broth [47] and incubated under anaerobic conditions at 37 °C for 48 h. Purity of the broth culture was checked by plating 50 μL of the Cardella broth in 2 BAB plates. The plates were then incubated in aerobic and anaerobic conditions, and only DNA extracted from pure *C. botulinum* group III cultures was subjected to the WGS. Genomic DNA was extracted from a 48-h broth culture using a Masterpure Gram-positive DNA purification kit (Epicentre, CA, USA) following the manufacturer’s instructions.

WGS was performed on the platforms NextSeq 500 (58 strains) and on MiSeq (2 strains) (Illumina, CA, USA). Libraries were prepared using the Nextera XT DNA kit. The High Output Kit v2 with paired-end 150-nt reads (300 cycles) was used for NextSeq 500, and the v3 Reagent kit was used for 600 cycles PE for MiSeq following the manufacturer’s instructions (Illumina, CA, USA).

### 2.3. Genome Assembly

The reads obtained from Illumina sequencing were trimmed using Sickle [48] with the following parameters: minimum quality score = 25 and minimum read length = 50 bps. The selected reads were assembled in contigs using SPAdes 3.7.0 [49]. The assembled genomes were submitted to the NIH genetic sequence database (Genbank—https://www.ncbi.nlm.nih.gov/genbank/ (accessed on 27 February 2020)) under the accession numbers reported in Appendix A.

### 2.4. Phylogenetic Analysis

The phylogenetic relations between genomes were analyzed using Gegenees software [50]. The program basics consist of the fragmentation of the analyzed sequences and an all-against-all comparison of the samples by BLASTN alignment [51] of the obtained fragments. The calculated average scores of all fragment comparisons is taken as a measurement of overall genomic similarity. For the whole genome analysis, the fragment length and step-size were both set at 500 bps, while for the BoNT-gene-carrying plasmid sequence analysis, the fragment length was set at 200 bps and step-size was set at 100 bps. To identify the contigs containing BoNT-gene-carrying plasmid sequences, each assembly was aligned against reference BoNT-gene-carrying plasmid sequences using the program MAUVE [52]. The references were BKT015925 (NC_015417), p1CbDC5 (JDRY01000001) and *Clostridium* phage c-st (NC_007581). The Gegenees results were exported as a similarity matrix in Nexus file format. The similarity matrix was used to construct dendrograms with the algorithm UPGMA using the online bioinformatic tool D-UPGMA (http://genomes.urv.cat/UPGMA/ (accessed on 28 November 2019)) [53].

When Gegenees phylogenetic analysis was unable to distinguish genomes within the same group, a SNP discovery analysis was performed with the program kSNP version 3 [54]. An ad hoc script was used to generate a difference matrix from the kSNP output files SNPs-all-matrix.fasta. Another ad hoc script was used to evaluate, by analysing the reads, if the found SNPs in the new sequenced genomes were real or artefacts as a product of the assembly process, and difference matrices were produced based only on the SNPs considered real. From the difference matrices, an UPGMA dendrogram was drawn using the online bioinformatic tool D-UPGMA (http://genomes.urv.cat/UPGMA/ (accessed on 28 November 2019)) [53].

All dendrograms were graphically elaborated by MEGA5 [55].

### 2.5. Bont and FliC Characterization

BoNT and *fliC* sequences were identified and extracted from each genome using BLAST [51] and then aligned with MEGA5 software [55].

### 2.6. Plasmidome Analysis

A contig was recognised as a circular molecule if an equal sequence, at least 70 bps long, was present at both ends and not found in other contigs of the same assembly. Contig self-adjacency was ascertained by analyzing the reads. The putative plasmids were confirmed using the program plasmidSPAdes [56].

The contigs considered as probably plasmids were compared among themselves and with representative sequences of the known *C. novyi sensu lato* plasmid categories using the Gegenees software (fragment length = 200 bps, step-size = 100 bps). To classify the contigs in plasmid categories (previously or newly established), the values regarding the contribution to the score, with a 40% threshold, have been considered [50].

The ORF search and translation in amino acid sequences were conducted using the program Unipro UGENE [57]. The obtained amino acid sequences were characterized with BLASTp searches [55] on the complete NBCI database.

The sharing of genomic regions between plasmid categories has been examined by BLAST searches [51] and MAUVE alignments [52].

The presence/absence of plasmid classes has been checked in genomes by BLAST searches [51] using as a subject from 1 to 3 reference sequences for each class and as query all the contigs of an assembly.

### 2.7. Recombination Analysis

To evaluate the homolog recombination in the *C. novyi sensu lato* group, the software ClonalFrame [58,59] was applied to a representative subset of the analyzed genomes. Pipeline and parameters used to reconstruct the clonal phylogeny and to estimate the recombination incidence are described in Giordani et al. [60].

### 2.8. Flagellin Locus Sequencing

The sequences of the four strains isolated in outbreak 1 (Table 1) were used as a template to design a set of seven couples of primers using Primer BLAST software. Before synthesis (performed by Eurofins MWJ, Ebersberg, Germany), oligonucleotide specificity was checked by in silico analysis against the published genomes on the Genbank database using BLAST software. PCR fragments were obtained using the primers listed in Table 2, and the NucleoSpin extract II kit (Macherey-Nagel, Düren, Germany) was used for fragment purification. Then, 10 ng of each fragment was sequenced using the same primers by Genechron (Rome, Italy). All the obtained sequences were aligned using CLC sequence viewer V6 (CLC bio, Aarhus, Denmark), and a consensus sequence corresponding to the *fliC* gene was determined.

Then, the resulting sequence was compared with other available flagellin gene types (*fliC*-I to -VI) performing both nucleotide and amino-acidic sequence alignments with MEGA5 [55] and CLC sequence viewer V6.

### 2.9. Nucleotide Sequence Accession Number

The GenBank accession numbers of new sequences of 60 Italian *C. botulinum* group III strains were reported in Table 1, while the two new *fliC* variants were deposited as following: IZSVeTV_10380 strain *fliC*-VI (accession number MT219975) and -VII (accession number MT219976).

## 3. Results

### 3.1. Sequenced Genomes

The genome of 60 *C. botulinum* group III strains was sequenced. The strains were isolated in Italy from biological (53) or environmental (7) samples collected in 32 animal botulism outbreaks. The affected animals belonged to different avian species (wild birds or commercial poultry) or bovines. The majority of the strains (52/60) were isolated in outbreaks which occurred in regions of the northeast of Italy (Veneto, Friuli Venezia Giulia, Trentino Alto Adige and Emilia Romagna); three occurred in regions of the northwest (Lombardia), four occurred in the center and one occurred in the south of Italy (Puglia) (Table 1). The sequencing raw output was de novo assembled. The depth coverage varied from 35× to 780×, and on average was 348×. The total length of the >500 bps long contigs ranges from 2,544,366 to 3,466,571 (Appendix A). Eleven strains resulted, carrying BoNT type C, 28 type C/D and 21 type D/C encoding genes (Table 1).

### 3.2. Genome Clustering

A phylogenetic analysis of the 60 newly sequenced genomes and of the already available genomes of *C. novyi sensu lato* was performed, and a dendrogram was created (Figure 1). Four main lineages, already described [16], were recognized (I, II, III, IV), with lineage I subdivided in two well-distinguished branches named IA and IB. Regarding the new sequenced genomes, 28 are located in lineage IA, 21 in lineage IB, 11 in lineage II, and none in lineage III and IV. Their serotype is congruent with the genome clustering: all the genomes in lineage IA are C/D, those in lineage IB are all D/C and those in lineage II are all C.

Regarding their geographical origin, strains from the north of Italy were found in two groups: 45 in lineage I (24 in IA, 21 in IB) and 11 in II; the four strains from central and southern Italy were all in lineage IA. All Italian lineage II strains were isolated in bovine botulism outbreaks which occurred in Bolzano province (in the Alps), with the exception of a unique strain isolated in the Rovigo province (Po Valley) (Figure 2).

In lineage IA (Figure 1), four main sub-branches can be distinguished: the largest of them (subcluster 1) gathers 30 strains of different geographical origin, including 14 isolated in Italy, 12 in France, 3 in Sweden (one of them is the complete genome BKT015925) and 1 in Spain. The other three sub-branches (subclusters 2, 3 and 4) include a few strains (5, 4 and 6 strains, respectively) all isolated in Italy except for strain 12LRNI13, which was isolated in France. The remaining strains, 49,511 from France and 16,868 from The Netherlands (the only strain of lineage IA that harbours the BoNT/D encoding gene) are singletons. All chimeric C/D strains of cluster I A were isolated from outbreaks which occurred in birds.

Additionally, lineage IB can be divided into four sub-branches: in the largest one (subcluster 1), 20 genomes are found, with 18 belonging to strains isolated in Italy and two in France; the second (subcluster 2) does not contain new genomes from this study, but includes 7 strains all isolated in the Southern hemisphere (5 in Brazil, 1 from the South African Republic, 1 in New Caledonia) [43,61]; the third (subcluster 3) contains 4 genomes isolated in Italy from the same outbreak; finally, the fourth subcluster contains 2 genomes isolated in Scandinavian countries (Denmark and Sweden). Cluster IB includes only BoNT/DC-producing strains isolated in bovine botulism outbreaks, with the exception of the unique genome of a strain isolated from a stork that resulted in the production of type D/C.

Lineage II includes *C. haemolyticum*, *C. novyi* and *C. botulinum* group III, producing only non-chimeric BoNTs (1 type D and 12 type C) isolated from mammals or environmental samples.

The lineage II *C. botulinum* branch can be further divided into three sub-branches (subclusters 1, 2, 4) and a strain which is a singleton (IZSVeTV/900/3/11, subcluster 3). The largest of them (subcluster 1) gathers 10 genomes: 9 from Italy and 1 from Sweden (Stockholm strain); the second branch (subcluster 2) includes 3 strains: 2 isolated in Italy and 1 in Africa (Chad).

For a more accurate evaluation of the genetic distances among very similar genomes, some sub-groups were analyzed with the program kSNP. From the analysis, it was evident that the genomes in these sub-groups, which were isolated in European countries which also had great distance between them, differ for a number of SNPs < 300 (Appendix A). For 17 epidemiological events, more than 1 isolated strain was sequenced (Table 1). Except for outbreaks with ID7 and ID9, the strains isolated from the same outbreak resulted in clones: 3 SNPs were identified in ID17, 2 SNPs in ID15, and 1 in ID5, 11 and 16, while no genetic differences were detected in the other events (Appendix A). Differently, in the events ID7 and ID9, the genetic distance (>1000 SNPs) means these isolates, despite being in the same outbreak, are considered different strains, though they are disposed in the same subcluster (subcluster 2 of lineage II and subcluster 2 of lineage IA, respectively) on a Gegenees dendrogram (Figure 2). Additionally, several genomes of strains isolated in different outbreaks of Italian origin are identical or differ for very few SNPs (<10) (Appendix A).

### 3.3. BoNT-Gene-Carrying Plasmid Clustering

The phylogenetic analysis of the BoNT-gene-carrying plasmid was obtained using Gegenees. The dendrogram representing the results is shown in Figure 3. The clustering substantially coincides with the phylogeny of the entire genome. The evident exceptions are the following: genome 49,511 of lineage IA and 4 IZSVeTV 10,380 (_13/14, _20/14, _18/14 and _17/14) of lineage IB appear in the same groups but not in the same sub-groups in the BoNT-gene-carrying plasmid dendrogram with respect to the genome dendrogram.

### 3.4. BoNT Gene Sequence Comparison

In the new genomes, three BoNT gene sequences were different from those already characterized; one was from a BoNT/DC strain and two from C. All obtained D/C sequences were equal to that of the DC5 genome, except one of the genome IZSVeTV_5262/1a/14, which differed for one nucleotide in position 3158, a missense mutation corresponding to an amino acid substitution from glycine to valine. All the C/D sequences are equal to BoNT gene sequences of BKT15925.

A total of 9 out of 12 sequences of the *C. botulinum* strains producing non-chimeric type C BoNT were shown to be identical to the genome of the Stockholm strain. The BoNT gene sequence of IZSVeTV_900/3/11 has 2 nucleotide substitutions with respect to the Stockholm genome in positions 1055 and 1675, while the sequence of IZSVeTV_7573/3/12 and IZSVeTV_7573/4/12 (equal between them) has 4 nucleotide substitutions, with 2 being the same as IZSVeTV_900/3/11 and the other 2 in positions 432 and 1023. Among these mutations, only 1675 is a missense mutation, implying a substitution from valine to isoleucine; all the others are silent.

No reduction in toxicity was observed during the mouse bioassay for the strains with one amino acid substitution.

### 3.5. Plasmidome Comparison

The genomes of most of the bacteria belonging to *C. novyi sensu lato* contain a large number of plasmids, of which many are probably circular prophages (ranging between three to five). The plasmids identified in *C. novyi sensu lato* have been classified into 13 different categories (PG1–PG13) [16]. The presence of sequences homologous to these 13 categories has been checked in the new sequenced genomes. The findings were as follows: PG1 (the BoNT-gene-carrying plasmid) and PG2 were present in all genomes; PG3 was found only in genomes of lineage I, except for two genomes of lineage IB; PG7 was present in many IA genomes, while PG6 and PG8 were present in some IA genomes; PG5 was present in all the II group genomes, while PG10 in many genomes of the same group (Appendix A).

Moreover, in the new sequenced genomes, circular DNA molecules have been recognized that did not match any of the previously described categories and have been classified into 10 new plasmid categories (PG14-23) (Appendix A). For each one of these categories, ORFs were searched. Part of the found ORFs showed homology with characterized genes, allowing some speculation on plasmid nature and properties (Appendix A). PG15 and PG17–PG22 contained several genes related to the viral life cycle and viral structure (terminase, integrase, phage tail tape measure protein, portal protein, head-tail adaptor protein, capsid protein, tail fiber protein encoding genes and others); therefore, they were likely non-integrated prophages. PG14 and PG16 possessed a recombinase, but did not possess other viral genes. Additionally, some genes not directly connected with the viral structure and physiology have been identified: transporter encoding genes (PG14 and PG23), beta-lactamase homolog genes (PG15), a bateriocin homolog gene (PG22), a nitroreductase homolog gene (PG19 and PG22), a probable sulfur metabolism gene (PG20), and others. An interesting feature, discovered in PG20, is an alpha toxin gene highly similar to *C. novyi* alpha toxin (95% nucleotide similarity) (Appendix A). It has been previously reported that in *C. novyi*, the alpha toxin-encoding genes are also located on plasmids that can be classified into categories PG10 and PG11 [16].

It has been observed that the four plasmid categories PG8–PG11 can be considered to form a cluster of plasmid groups on the basis of the evident sharing of genetic content, likely a consequence of phage module exchange [16]. The shared genetic material appeared to be organized in three modules: integration and recombination, structural phage genes and a lysis-lysogeny-regulating module. Additionally, PG20 contained regions, in addition to the alpha toxin gene, that manifested homology with some of the shared regions in cluster PG8–PG11. In particular, PG20 shares a 2046 bps long region with PG10 plasmids (p1CnGD211209NZ-a.n. CM003331, p1CnBKT29909-a.n. NZ_CM003328 and p1Cn4540-a.n. NZ_CM003326), and with p1CnGD211209, it shares a 19,618 bps long region corresponding to the structural phage gene module of this plasmid (the nucleotide similarity is around 80–85%) (Appendix A).

PG14 contained some regions also present in PG7, with a total length of about 5 Mb and a nucleotide similarity ranging from 88% to 96%. These regions comprised the CDSs of p5BKT015925 (the plasmid of class PG7 belonging to the reference genome BKT015925-a.n. NC_015419) CBC4_RS15070, CBC4_RS15100, CBC4_RS15105, CBC4_RS15120, and CBC4_RS15125 (Appendix A).

The presence or absence of these newly discovered plasmid was specific for the different *C. novyi sensu lato* clusters (Appendix A): PG15, PG18, PG21 and PG22 are found in cluster IA, PG14, PG16 and PG20 are found in cluster IB and PG17, PG19 and PG23 are found in cluster II.

In some case, strains isolated in the same outbreak that were shown to be equal by SNP analysis do not appear to contain the same plasmids (Appendix A). This can be due to the low stability of the plasmids through cell division, but could also be the mere effect of plasmid loss during DNA extraction or gaps in sequencing.

### 3.6. Recombination Analysis

A total of 20 out of 60 analyzed genomes were selected as representative of the *C. novyi sensu lato* clusters and subclusters to check the recombination rate in this taxon by using the program ClonalFrame. As result of the ClonalFrame run, the calculated r/m ratio is 0.9. The r/m ratio is considered a marker of the recombination frequency. According to the criteria proposed by Vos and Didelot [62], 0.9 is a low r/m ratio, suggesting an infrequent recombination. The clonal phylogeny, produced by ClonalFrame (data not shown), is coherent with the topology of the dendrogram obtained with Gegenees (Figure 1).

### 3.7. fliC Analysis

To confirm the genetic variability of *fliC* and its reliability as a genetic marker, *fliC* gene sequences obtained from newly sequenced genomes were investigated during this study. Among the 60 Italian samples, 28 strains were typed as *fliC*-I, 17 as *fliC*-IV and 11 strains as *fliC*-II. Our data showed that all 28 *fliC*-I strains were C/D and all *fliC*-II strains were C. Instead, 16 out of 20 D/C strains harboured *fliC*-IV. None of the sequenced strains were found to be positive for *fliC*-III or *fliC*-V.

Four strains isolated in outbreak 1 did not show any of the already known *fliC* variants. Data were also confirmed by Sanger sequencing.

The sequencing showed, in these genomes, the presence of two *fliC* genes, in adjacent genomic position, with a different length: 1221 bps and 1083 bps, respectively. The % AA similarity with other *fliC* types is, for the first one, around 50 (max. similarity 57.11 with *fliC*-IV), and, for the other one, around 60 (max. similarity 61.93 with *fliC*-III). The % AA similarity of the two *fliC* genes with each other is 82.8. The two sequences were considered two new *fliC* types and called *fliC*-VI and *fliC*-VII (Table 3 and Figure 4).

Analyzing the sequence of *fliC*-I, -II, -III and -V, their higher degree of similarity (84–92%) with respect to *fliC*-IV, -VI and -VII (47–62%) becomes evident (Table 3 and Figure 4).

Fragments of *fliC* type VI and VII sequences were also found in two previously sequenced genomes, BKT2873 and BKT75002, both originating in Sweden, with 100% similarity.

## 4. Discussion

In the last decade, the availability of next generation sequencing (NGS) technologies has made it possible to produce whole genome sequences of a large number of bacteria in a short time. As a consequence, genotyping studies have benefited from a technique that allows a considerably greater discrimination and accuracy power than the standard genotyping methods (PFGE, AFLP, MLVA, MLST) based on a limited selection of the genetic elements.

NGS has also been applied to *C. botulinum* group I [60,63], group II [64], and to a restricted number of group III genomes [16,43,63]. There are three published NGS studies of *C. botulinum* group III. In Skarin et al. [16], the genomes of 13 *C. botulinum* group III and 11 other *C. novyi sensu lato* strains were analyzed. These were prevalently isolated in Sweden, but also in other European countries (The Netherland, Spain, Denmark and 2 strains in Italy) and worldwide (Chad, South African Republic). Some were isolated from recent outbreaks and others were received from old collections, isolated more than 50 years ago. In Woudstra et al. [43], 17 *C. botulinum* group III genomes were sequenced and compared with those already available, 16 of which were isolated in France in the period 2008–2013, and 1 which was isolated in New Caledonia (Oceania) in 2013. In addition, Woudstra and coworkers (2017) reported the draft genome sequence of five Brazilian *C. botulinum* Group III type D/C strains [61].

The restricted number of *C. botulinum* group III bacteria included in the above-mentioned studies is probably due to the difficulties in the isolation of those microorganisms. Indeed, the isolation of *C. botulinum* group III strains requires (i) cultivation and repeated transfers in complex growth media, (ii) strict anaerobic conditions and (iii) the preservation of the neurotoxin encoding gene, which is the target for the definitive biomolecular identification of the isolate. The latter requirement is probably the most challenging due to the unstable lysogeny of the phage carrying the genes encoding for BoNT type C, D, C/D, and D/C, which are easily lost under laboratory conditions or techniques [15,23,65].

In the present study, we sequenced the whole genome of 60 *C. botulinum* group III strains isolated from biological and environmental samples collected in animal botulism outbreaks, and we compared the new sequences with already available ones. Our study, confirming many findings and derived hypotheses exposed in the previous studies, adds new information concerning the genetic picture of the European *C. botulinum* group III strains.

Four lineages have been characterized in *C. novyi sensu lato*, and it has been hypothesized that these lineages represent the main population of this group or, at least, of the strains able to cause botulism in mammals and birds [16]. All the new genomes sequenced during this study are unequivocally comprised of one of these lineages.

It has been postulated that one clade of closely related strains, responsible for avian botulism, dominates in Europe [25,43]. This hypothesis was supported by the increase of avian botulism outbreaks in poultry production observed in France between 2007 and 2011 [43]. The present data substantially confirm this observation, even though the diffusion of other clades, at least in Italy, is relevant. Indeed, 28 out of 60 new genomes are included in cluster IA, of which 14 are very similar to the genome BKT015925, forming a sub-branch that contains a large part of the strains isolated in Europe that produce BoNT type C/D and are responsible for botulism outbreaks occurring both in poultry and in wild birds. The fact that *C. botulinum* group III harbored by wild birds was genetically strictly related to strains responsible for avian botulism of poultry suggests the respect of severe biosecurity rules at the farm level to avoid not only the introduction of very impactful diseases such as avian influenza, but also the occurrence of avian botulism caused by strains spread to commercial flocks from wild birds. Interestingly, the unique strains included in lineage IA not related to avian species/environment harbor a BoNT/D encoding gene and show the lowest homology with the other strains encompassed in this lineage. To clarify, if one clade of strains responsible for avian botulism dominates only in Europe, the genome sequence of strains isolated in other parts of the world should be compared. This study revealed the existence of new sub-branches in lineage IA that include only Italian strains, but this could be modified in the future with the increase of the genomes of *C. botulinum* group III originating in different countries worldwide.

Additionally, in lineages IB and II, some sub-branches appear predominant with respect to the others: in lineage IB, the majority of strains are very similar to the genome DC5; in lineage II, a group of genomes very similar to Stockholm (8 of 11) constitutes the prevalent sub-branch, for which a pan-European distribution can be supposed, since the strains it is comprised of were isolated in countries (Sweden and Italy) at the northern and southern tips of the continent. Lineage II gathers many strains new to this study that were also isolated from 2011 to 2013, so the observation, reported in Skarin et al. [16], that this lineage was more common in past decades and rare today should be revised.

A large number of plasmids or circular prophages were discovered in strains of the *C. novyi sensu lato* group. It was hypothesized that these plasmids are part of a larger pan-plasmidome, and medium and small plasmids (in particular bacteriophages) are the major vehicle for lateral gene transfer in this taxon through the exchange of genetic material between plasmids and lateral mobility between strains. It has been proposed that, by this mechanism, several virulence genes, such as the alpha toxin and beta toxin gene, have been exchanged among strains [16,42]. In the genomes sequenced in this study, several plasmids have been discovered. Some of them can be classified into one of the 13 categories already proposed, while others constituted 10 new categories that widen the known pan-plasmidome catalog of *C. novyi sensu lato*. On the basis of the gene content, many of the new categories seem to be composed of prophages. Two new categories show evidences of gene exchange with other plasmid categories: PG14 and PG20 share genetic material, respectively, with PG10 (found in genomes of lineage II, III and IV) and PG7 (present in genomes of lineage IA). PG14 and PG20 are both retrieved in strains of lineage IB, which constitutes an example of probable gene transfer between lineages. In relation to virulence factor transfer, in PG20, the presence of an alpha toxin gene (very similar to that of lineage IV *C. novyi*) is noteworthy. This finding suggests the hypothesis that PG20 can be the vehicle for an alpha toxin gene to enter the IB lineage. Differently from the small plasmids, the big-sized plasmids, in particular the BoNT-gene-carrying plasmid, have shown to be unfrequently transferred by horizontal heredity mechanisms, since the phylogeny built on their sequences is the same as the whole genome sequence-deduced phylogeny [16,42]. This observation is substantially confirmed by the present analysis, as it can be seen when comparing the whole genome dendrogram (Figure 1) and the BoNT-gene-carrying plasmid dendrogram (Figure 3), even if slight differences in topology between the two dendrograms do not allow the possibility of BoNT-gene-carrying plasmid lateral exchange to be excluded.

It has been observed that the mosaic C/D BoNT serotype is associated with avian bot-ulism, while recombinant D/C and C serotypes are associated with mammalian botulism. The only *C. botulinum* type D/C isolated in a non-mammal species was detected in an avian species (stork) that is carnivorous and may also consume small mammals.

The BoNT sequence within the same serotypes (D/C, C/D and C) was very conserved: only three sequences differ from the others of the same serotype, and for no more than four nucleotides. Therefore, according to the criteria suggested by Peck et al. [66] for the definition of subtypes among types A, B, E and F, no new subtypes can be pointed out among the BoNT encoding genes sequenced in this study.

*C. botulinum* group I and II are well known to be highly diverse in their genomic features, and flagellin locus also shows this trait. The flagellin variable region (flaVR) in groups I and II shows 15 different variants [46]. Otherwise, strains belonging to group III are genetically less heterogeneous, and only 7 flagellin gene (*fliC*-I–VII) variants were described, comprising the two new variants discovered during this study. *fliC*-I was the most common type linked to the C/D serotype, followed by type IV, which was linked to the D/C toxin. None of the 60 analyzed strains were found to contain *fliC*-III or *fliC*-V, confirming that these *fliC* types are very rare, at least in our geographical region.

In four of the new sequenced genomes, two new *fliC* variants were discovered: *fliC*-VI and *fliC*-VII, both of which were present in tandem. The presence of different *fliC* types in the same genome could be the result of two different horizontal gene transfer events.

On the basis of the ClonalFrame results, homologous recombination in the *C. novyi sensu lato* seems to be rare, in contrast with *C. botulinum* group I, where this recombination has been shown to be relatively frequent [60]. This is not surprising, considering that the two taxons are genetically and biologically distinct and distantly related.

The results obtained in this study also allows for epidemiological speculations about botulism in cattle and poultry. Indeed, this is the first study in which the genomes of different strains isolated in the same outbreak have been compared. These strains were isolated from different subjects or from environmental samples suspected to be the source of the disease. To clarify some epidemiological aspects of bovine and avian botulism, the genomes of the strains isolated in the same outbreaks where deeply analyzed by means of kSNP software. The results revealed that, in bovine botulism outbreaks, the strains isolated from different subjects or from the source suspected to be the carrier of the strain (feed contaminated by animal carcasses or water) were clones or genetically strictly related. This observation supports the toxico-infective pathogenesis of bovine botulism, but does not exclude the simultaneous assumption of preformed neurotoxin (intoxication form). In outbreaks of bovine botulism, the feed samples were residues of the total mixed ration (TMR), collected from the bottom of the mixer wagon a few days after the appearance of the disease.

This study suggests some practical information that can be taken into consideration to prevent botulism in poultry. The results confirm that rice hulls used for poultry litter is at risk for the introduction of *C. botulinum group* III strains, as previously speculated [24]. Indeed, in one meat chicken outbreak, the strains isolated from two dead birds (2559/1/10 and 2559/2/10) and the strain (4879/20/10) isolated from the rice hull batch used to prepare the litters showed a clonal origin. In one different outbreak which occurred in meat chickens, the strains isolated from the birds and the rice hulls appeared to be different, but the risk of this kind of litter for poultry still remains. For this reason, rice hulls should be guaranteed as *C. botulinum*-free by the producers.

The results also clarify some epidemiological aspects of botulism that occur in pheasants. These birds are usually bred in a semi-wild environment in large aviaries with many bushes. In this condition, it is very difficult to find and quickly remove the carcasses of birds that died because of botulism. This problem eventually increases the rate of contamination of the soil due to the accumulation of spores originating from the spoils of dead birds over the years. The close genetic similarity of the genomes of a strain isolated from a dead pheasant (6503/1/13) and of a strain isolated from the soil (7494/7/14) of its aviary supports this hypothesis.

## 5. Conclusions

This study confirms that the sequence of the neurotoxin gene of the four BoNT serotypes (C, D, C/D and D/C) produced by *C. botulinum* group III are very conserved, contrary to what happens for BoNT type A, B, E and F, and no new subtypes have been detected.

Moreover, it consolidates the existence of five main lineages (IA, IB, II, III, IV) in *C. novyi sensu latu* and reveals the existence of new *fliC* types and new plasmids.

WGS showed to be a very promising method to investigate the epidemiology of animal botulism.

The genomic analysis of many strains of *C. botulinum* group III, isolated in recent years in outbreaks occurring in Italy, has been profitable for the expansion of knowledge on the genetic diversity in this group and its geographical distribution. For a complete description, it is still necessary to sequence and study the genome of this group of strains coming from different regions of the world.

## Figures and Tables

**Figure 1 microorganisms-09-02347-f001:**
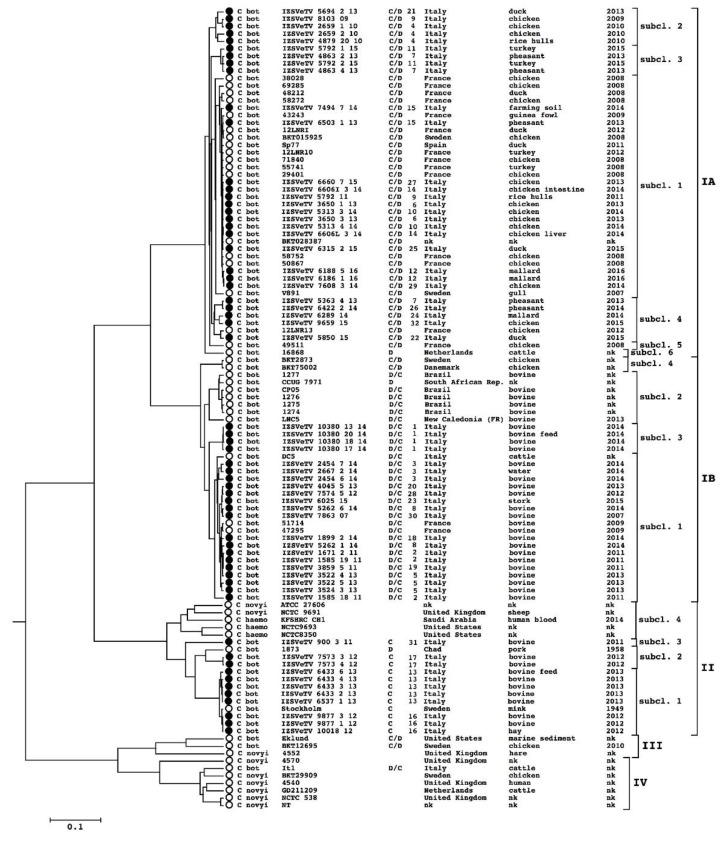
Dendrogram representing the phylogenetic relations among the analyzed strains, as calculated using Gegenees software. The new sequenced genomes are marked with black dots while the genomes already analyzed in previous studies are marked with white dots. For each sample, the information shown in the Figure are, from left to right: species, sample name, serotype, outbreak ID, geographical origin, isolation source, isolation year.

**Figure 2 microorganisms-09-02347-f002:**
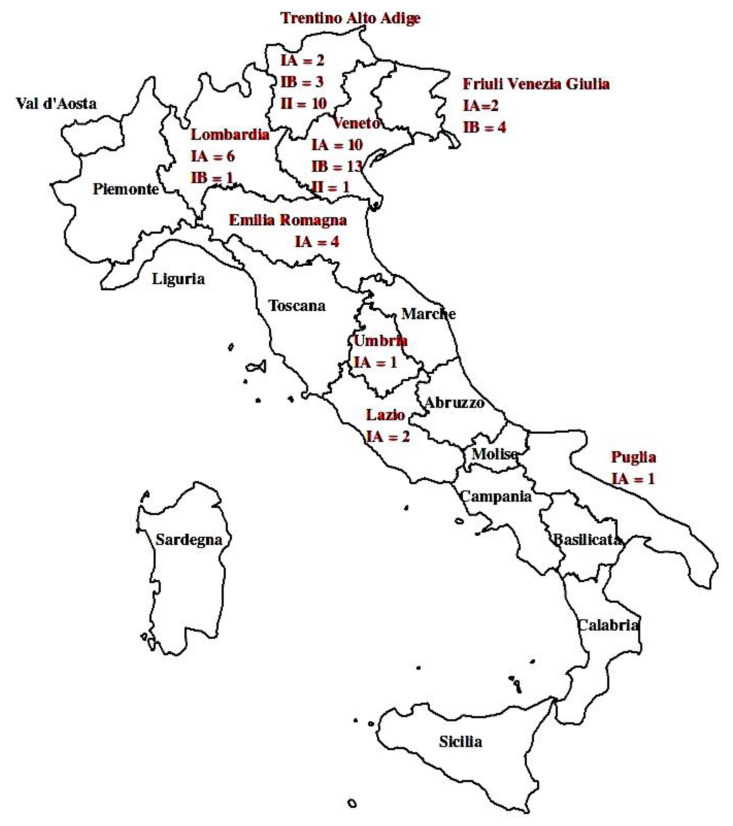
Geographical origin of the new sequenced strains: the numbers shown represent the strains isolated in each Italian region. The names of the regions where strains were isolated are in red.

**Figure 3 microorganisms-09-02347-f003:**
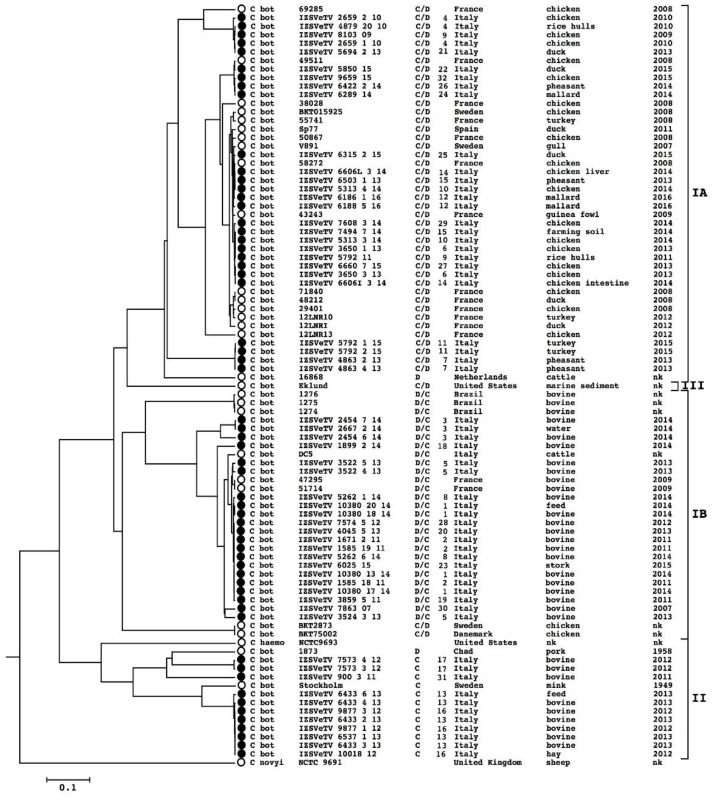
Dendrogram representing the phylogenetic relations among the BoNT-gene-carrying plasmid sequences of the analyzed strains, as calculated using Gegenees software. The new sequenced genomes are marked with black dots, while the genomes already analyzed in previous studies are marked with white dots. The group designations are the same used for the genome sequence classification. For each sample, the information shown in the Figure are, from left to right: species, sample name, serotype, outbreak ID, geographical origin, isolation source, isolation year.

**Figure 4 microorganisms-09-02347-f004:**
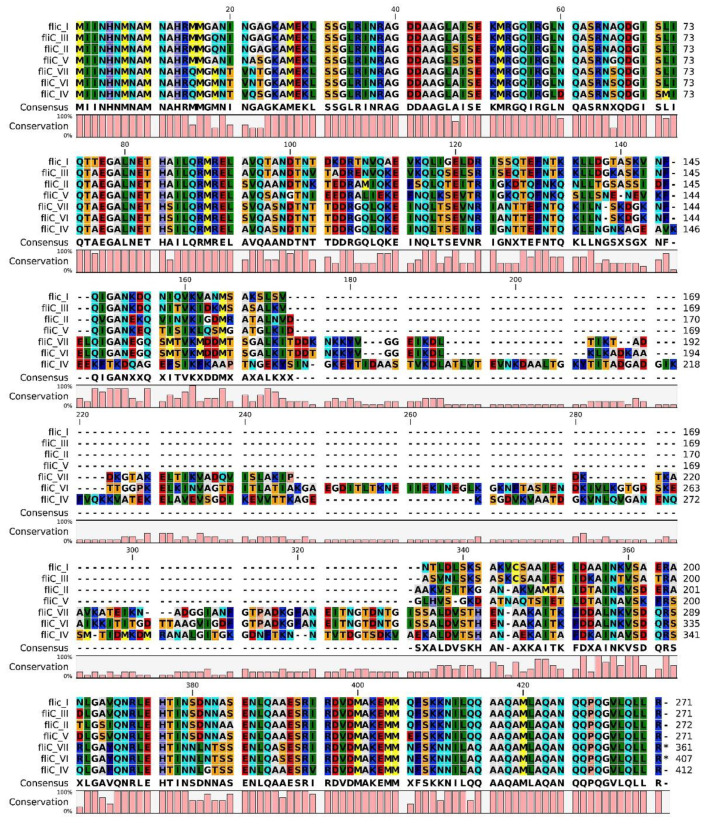
Alignment of the *fliC* type aminoacidic sequences.

**Table 1 microorganisms-09-02347-t001:** Information on the new sequenced strains analyzed in the present study.

Sample Name	Serotype	*fliC* Type	Collection Year	Collection Location	Isolation Source	Host	Outbreak ID
IZSVe-TV 10380/13/14	D/C	VI-VII	2014	Cervignano del Friuli (UD)–Friuli VG	faeces	*Bos taurus*	1
IZSVe-TV 10380/17/14	D/C	VI-VII	2014	Cervignano del Friuli (UD)–Friuli VG	rectum	*Bos taurus*	1
IZSVe-TV 10380/18/14	D/C	VI-VII	2014	Cervignano del Friuli (UD)–Friuli VG	colon	*Bos taurus*	1
IZSVe-TV 10380/20/14	D/C	VI-VII	2014	Cervignano del Friuli (UD)–Friuli VG	feed		1
IZSVe-TV 1585/18/11	D/C	IV	2011	Piove di Sacco (PD)–Veneto	ruminal content	*Bos taurus*	2
IZSVe-TV 1585/19/11	D/C	IV	2011	Piove di Sacco (PD)–Veneto	ruminal content	*Bos taurus*	2
IZSVe-TV 1671/2/11	D/C	IV	2011	Piove di Sacco (PD)–Veneto	ruminal content	*Bos taurus*	2
IZSVe-TV 2454/6/14	D/C	IV	2014	Mansuè (TV)–Veneto	faeces	*Bos taurus*	3
IZSVe-TV 2454/7/14	D/C	IV	2014	Mansuè (TV)–Veneto	faeces	*Bos taurus*	3
IZSVe-TV 2667/2/14	D/C	IV	2014	Mansuè (TV)–Veneto	drinking water		3
IZSVe-TV 2659/1/10	C/D	I	2010	Calvenzano (BG)–Lombardia	intestine content	*Gallus gallus*	4
IZSVe-TV 2659/2/10	C/D	I	2010	Calvenzano (BG)–Lombardia	intestine content	*Gallus gallus*	4
IZSVe-TV 4879B/20/10	C/D	I	2010	Calvenzano (BG)–Lombardia	rice hulls (litter for chicken)		4
IZSVe-TV 3522/4/13	D/C	IV	2013	Trebaseleghe (PD)–Veneto	faeces	*Bos taurus*	5
IZSVe-TV 3522/5/13	D/C	IV	2013	Trebaseleghe (PD)–Veneto	faeces	*Bos taurus*	5
IZSVe-TV 3524/3/13	D/C	IV	2013	Trebaseleghe (PD)–Veneto	intestine content	*Bos taurus*	5
IZSVe-TV 3650/1/13	C/D	I	2013	Caprino Veronese (VR)–Veneto	liver	*Gallus gallus*	6
IZSVe-TV 3650/3/13	C/D	I	2013	Caprino Veronese (VR)–Veneto	intestine content	*Gallus gallus*	6
IZSVe-TV 4863/2/13	C/D	I	2013	Castelvetro (PC)–Emilia Romagna	gastric content	*Phasianus colchicus*	7
IZSVe-TV 4863/4/13	C/D	I	2013	Castelvetro (PC)–Emilia Romagna	gastric content	*Phasianus colchicus*	7
IZSVe-TV 5363/4/13	C/D	I	2013	Castelvetro (PC)–Emilia Romagna	intestine content	*Phasianus colchicus*	7
IZSVe-TV 5262/1/14	D/C	IV	2014	San Zenone degli Ezzelini (TV)–Veneto	faeces	*Bos taurus*	8
IZSVe-TV 5262/6/14	D/C	IV	2014	San Zenone degli Ezzelini (TV)–Veneto	faeces	*Bos taurus*	8
IZSVe-TV 5792/11	C/D	I	2011	Volta Mantovana (MN)–Lombardia	rice hulls (litter for chicken)		9
IZSVe-TV 8103/09	C/D	I	2009	Volta Mantovana (MN)–Lombardia	intestine content	*Gallus gallus*	9
IZSVe-TV 5313/3/14	C/D	I	2014	Agugliaro (VI)–Veneto	intestine content	*Gallus gallus*	10
IZSVe-TV 5313/4/14	C/D	I	2014	Agugliaro (VI)–Veneto	intestine content	*Gallus gallus*	10
IZSVe-TV 5792/1/15	C/D	I	2015	Castelnovo Bariano (RO)–Veneto	liver	*Meleagris gallopavo*	11
IZSVe-TV 5792/2/15	C/D	I	2015	Castelnovo Bariano (RO)–Veneto	intestine content	*Meleagris gallopavo*	11
IZSVe-TV 6186/1/16	C/D	I	2016	Muggia (TS)–Friuli VG	intestine content	*Anas platyrhynchos*	12
IZSVe-TV 6188/5/16	C/D	I	2016	Muggia (TS)–Friuli VG	intestine content	*Anas platyrhynchos*	12
IZSVe-TV 6433/2/13	C	II	2013	Sarentino (BZ)–Trentino AA	faeces	*Bos taurus*	13
IZSVe-TV 6433/3/13	C	II	2013	Sarentino (BZ)–Trentino AA	faeces	*Bos taurus*	13
IZSVe-TV 6433/4/13	C	II	2013	Sarentino (BZ)–Trentino AA	faeces	*Bos taurus*	13
IZSVe-TV 6433/6/13	C	II	2013	Sarentino (BZ)–Trentino AA	feed		13
IZSVe-TV 6537/1/13	C	II	2013	Sarentino (BZ)–Trentino AA	ruminal content	*Bos taurus*	13
IZSVe-TV 6606L/3/14	C/D	I	2014	Naturno (BZ)–Trentino AA	liver	*gallus gallus*	14
IZSVe-TV 6606I/3/14	C/D	I	2014	Naturno (BZ)–Trentino AA	intestine content	*gallus gallus*	14
IZSVe-TV 6503/1/13	C/D	I	2013	Montalto di Castro (VT)–Lazio	liver	*Phasianus colchicus*	15
IZSVe-TV 7494/7/14	C/D	I	2014	Montalto di Castro (VT)–Lazio	Soil (pheasant farm)		15
IZSVe-TV 10018/12	C	II	2012	Scena (BZ)–Trentino AA	hay		16
IZSVe-TV 9877/1/12	C	II	2012	Scena (BZ)–Trentino AA	faeces	*Bos taurus*	16
IZSVe-TV 9877/3/12	C	II	2012	Scena (BZ)–Trentino AA	faeces	*Bos taurus*	16
IZSVe-TV 7573/3/12	C	II	2012	San Lorenzo di Sebato (BZ)–Trentino AA	faeces	*Bos taurus*	17
IZSVe-TV 7573/4/12	C	II	2012	San Lorenzo di Sebato (BZ)–Trentino AA	faeces	*Bos taurus*	17
IZSVe-TV 1899/2/14	D/C	IV	2014	Levico Terme (TN)–Trentino AA	bovine ruminal content	*Bos taurus*	18
IZSVe-TV 3859/5/11	D/C	IV	2011	Galgagnano (MI)–Lombardia	faeces	*Bos taurus*	19
IZSVe-TV 4045/5/13	D/C	IV	2013	Bleggio Superiore (TN)–Trentino AA	liver	*Bos taurus*	20
IZSVe-TV 5694/2/13	C/D	I	2013	Mogliano Veneto (TV)–Veneto	intestine content	*Anas platyrhynchos*	21
IZSVe-TV 5850/15	C/D	I	2015	Correzzola (PD)–Veneto	intestine content	*Anas platyrhynchos*	22
IZSVe-TV 6025/15	D/C	IV	2015	Quinto di Treviso (TV)–Veneto	intestine content	*Ciconia ciconia*	23
IZSVe-TV 6289/14	C/D	I	2014	Lesina (FG)–Puglia	intestine content	*Anas platyrhynchos*	24
IZSVe-TV 6315/2/15	C/D	I	2015	Perugia (PG)–Umbria	intestine content	*Anas platyrhynchos*	25
IZSVe-TV 6422/2/14	C/D	I	2014	Cremona (CR)–Lombardia	intestine content	*Phasianus colchicus*	26
IZSVe-TV 6660/7/15	C/D	I	2015	San Martino Buonalbergo (VR)–Veneto	intestine content	*gallus gallus*	27
IZSVe-TV 7574/5/12	D/C	IV	2012	Stenico (TN)–Trentino AA	faeces	*Bos taurus*	28
IZSVe-TV 7608/3/14	C/D	I	2014	Forlì (FC)–Emilia Romagna	intestine content	*Gallus gallus*	29
IZSVe-TV 7863/07	D/C	IV	2007	Colle Umberto (TV)–Veneto	faeces	*Gallus gallus*	30
IZSVe-TV 900/3/11	C	II	2011	Adria (RO)–Veneto	liver	*Bos taurus*	31
IZSVe-TV 9659/15	C/D	I	2015	Polesella (RO)–Veneto	intestine content	*Gallus gallus*	32

**Table 2 microorganisms-09-02347-t002:** Primers for *fliC* gene amplification.

Primer Name	Sequence 5′ → 3′	Amplicon Size (bp)
*fliC*_VI_F1_f	TGCTGAAAAAGCAAGAATACAGAAAAG	805
*fliC*_VI_F1_r	TGTTGCTGCTTTATCTGCCTTTAAT	
*fliC*_VI_F2_f	AGGAGCTAATGAAGGTCAATCAATG	907
*fliC*_VI_F2_r	CCTTTTCTCACTTCTTTGAATGGGT	
*fliC*_VI_F3_f	CGATTTAGTTAATGGAACTGAATCCAC	806
*fliC*_VI_F3_r	TAAGGCACCTGAAGTCATATCATCC	
*fliC*_VI_F4_f	CAGAAGTTAATAGAATAGCTAATACAACTG	981
*fliC*_VI_F4_r	GCCAGAGATTTATCTCCGACTCTTA	
*fliC*_VI_F5_f	AGAGGACAAATCAGAGGTCTTAACC	686
*fliC*_VI_F5_r	CCAATTACTCCAGCTGCAGTAGT	
*fliC*_VI_F6_f	AGAGGACAAATCAGAGGTCTTAACC	663
*fliC*_VI_F6_r	GTCAGCATTCTTTATTTCAGTGGCT	
*fliC*_VI_F7_f	AATCAACAACCACAAGGAGTTCTTC	539
*fliC*_VI_F7_r	GGTTAAGACCTCTGATTTGTCCTCT	

**Table 3 microorganisms-09-02347-t003:** Percent similarity matrix of the *fliC* type aminoacidic sequences.

*fliC* Variants						
*fliC*-I	/	92.66	84.86	83.94	61.24	50.69
*fliC*-III	92.66	/	86.93	84.86	61.93	51.38
*fliC*-II	84.86	86.93	/	86.01	61.47	50.92
*fliC*-V	83.94	84.86	86.01	/	59.86	49.31
*fliC* 10380 VII	61.24	61.93	61.47	59.86	/	82.8
*fliC* 10380 VI	50.69	51.38	50.92	49.31	82.8	/
*fliC*-IV	46.79	47.48	47.48	47.25	60.78	57.11

## Data Availability

All the genomes sequenced in the present study were deposited in the NIH genetic sequence database (Genbank—https://www.ncbi.nlm.nih.gov/genbank/ (accessed on 27 Ferbuary 2020) and will be publicly available once the study is published.

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
