# Peer review of "Extensive Genome Exploration of Clostridium botulinum Group III Field Strains"

_microorganisms, 2021, doi:10.3390/microorganisms9112347_

Round 1
Reviewer 1 Report
The paper describes the WGS of 60 strains of C. novyi sensu lato from Italian botulism outbreaks spanning nearly 10 years in the context of previously sequenced strains. The authors have described is sufficient detail the procedures followed and the data are well represented in general.
Novel findings have included the identification of 10 new classes of putative extra-chromosomal molecules, which is significant, and two novel sequence types of the flagellin gene fliC. The findings of the genome comparisons have generally confirmed previous findings. The information gained through this study has also elucidated some strategies to prevent future outbreaks.
The manuscript requires further editing, with the removal of un-necessary hyphenation, which has been unintentional but was very distracting whilst reading.
Specific comments:
Line 126 There is a 2007 strain…
Figure 3 this alignment is of the BoNT encoding plasmid like sequences? This should be stated in the figure legend. Likewise the group designations are that of the genome sequence, this should also be stated in the figure legend.
Table 4 removed the numbers 1-7 and just use the names of the fliC groups both vertically and horizontally. Why are they not in order – fliC I - fliC VII?
Could the authors use a . instead of a , to delineate the decimal point?
In supplementary table S3, sheet 2, could the plasmid group and strain names also be added to the other side of the matrix table as it is very large and this would ease comparison of the later groups.
Author Response
FIRST REVIEWER
The paper describes the WGS of 60 strains of C. novyi sensu lato from Italian botulism outbreaks spanning nearly 10 years in the context of previously sequenced strains. The authors have described is sufficient detail the procedures followed and the data are well represented in general.
Novel findings have included the identification of 10 new classes of putative extra-chromosomal molecules, which is significant, and two novel sequence types of the flagellin gene fliC. The findings of the genome comparisons have generally confirmed previous findings. The information gained through this study has also elucidated some strategies to prevent future outbreaks.
The manuscript requires further editing, with the removal of unnecessary hyphenation, which has been unintentional but was very distracting whilst reading.
R: the unnecessary hyphenation has been removed.
Specific comments:
Line 126 There is a 2007 strain…
R: the study temporal interval has been changed to include 2007 strain: “from 2007 to 2016 ”
Figure 3 this alignment is of the BoNT encoding plasmid like sequences? This should be stated in the figure legend. Likewise the group designations are that of the genome sequence, this should also be stated in the figure legend.
R: the legend of Figure 3 has been rewritten as follows: “Dendrogram representing the phylogenetic relations among the BoNT-gene-carrying plasmid sequences of the analyzed strains, as calculed using Gegenees software. The new sequenced genomes are marked with black dots while the genomes already analyzed in previous studies with white dots. The group designations are the same used for the the genome sequence classification. For each samples, the information shown in the Figure are, from left to right: species, sample name, serotype, outbreak ID, geographical origin, isolation source, isolation year.”
Table 4 removed the numbers 1-7 and just use the names of the fliC groups both vertically and horizontally. Why are they not in order – fliC I - fliC VII?
R: Table 4 has been rearranged as requested. The fliC types are listed in order of similarity.
Could the authors use a . instead of a , to delineate the decimal point?
R: the decimal numbers have been corrected according the requested convention.
In supplementary table S3, sheet 2, could the plasmid group and strain names also be added to the other side of the matrix table as it is very large and this would ease comparison of the later groups.
R: The table has been rearranged as requested.
Reviewer 2 Report
This study presents the results of sequencing of 60 C. botulinum group III isolated in Italy from 32 animal botulism outbreaks between 2008 and 2016. Before this study, only 58 genomes of C. novyi sensu lato were available in public databases (48 included in this study), demonstrating the great relevance and interest of this study by doubling the number of available genomes. The analysis of these genomes allowed the confirmation of the existence of 4 lineages in the C. novyi sensu lato group, of a large plasmidome and the relevance of the use of FliC genes for subtyping. New plasmids are described and two new FliC genes have been sequenced and fully compared to other previously described FliC genes. Moreover this study provides very useful information about epidemiology of animal botulism through source identification (Rice hulls for example), or by highlighting the risks of cross-contamination between wild birds and poultry as closely related strains are isolated in both cases.
The article is well written (minor typing errors) and easy to read.
Major comments:
Materials and methods:
Forty-eight genomes were included in this study. According to Le Gratiet et al. 2021, 58 genomes of C. novyi sensu lato are currently available. Moreover complete genomes of 4 strains were published in 2021 (Woudstra et al. 2021). Could authors please indicate when the genomes were retrieved here for their study so as to justify why the 10 other available genomes were not included in their study (line 142-143)? Citation of these two recent articles is not necessary (at least here) but to my point of view it is important to explain why all genomes that are currently available in public databases were not included.
Line 152-153: could authors please explain how they determine that culture was at the exponential growth phase (enumeration? Optical density? Other way?). How long was the incubation on average?
Results
Line 239-242: Most of the strains were isolated from the North-East of Italy. Is it because most animal outbreaks occurred in the region in Italy? Could please authors explain their choice about the distribution among the different regions? Why did they mostly sequence strains from this region and not select the same number of strains per region so as to have an equal distribution of the strains across the country? (This may be part of the discussion or in the MM section (2.1 strains) and not included in the results section)
Figure 1: could authors please add the reference of each outbreak in the dendrogram? The numbering is confusing, For example, the strain 3859/5/11 is part of outbreak 19 and not from outbreak 11. Addition of the outbreak number in the dendrogram will be very helpful for the reader.
Is it also possible to improve the quality of the Figure 1 and Figure 3 (they are blurry when zoomed in and numbers are difficult to read)?
Could authors please explain how they determined the subclusters within each lineage? Which criteria were used to distribute the strains in the different subclusters?
Table 2: is Table 2 really necessary in the article? Couldn’t it be considered as supplemental data?
Flagellin: could it be possible to add the FliC group of each strain in one Table or Figure? (Table 1 for example? Or in Figure 1?)
SNPs analysis:
Table S2: How should the data in these tables be interpreted? Is this the number of SNPs that differs between two genomes or the % of difference between two strains? This is not detailed neither in the legend in the article nor above the tables.
Line 286-288 and line 294 and line 298: Could authors please explain why they say that 300 SNPs is low and above 1000 is distant? What are the criteria to determine when the number of SNP is low or high?
Line 290-298: Regarding these results, some discussion is missing. Authors report between 1 and 3 SNPs for genomes of outbreaks 5, 11, 16 and 17. How should the presence of these SNPs be interpreted? Could these genomes be considered to be the same? Where are the SNPs located?
Line 448-449 : this has also been mentioned in Anza et al. 2014 where strains from Spain and From Sweden were compared (http://dx.doi.org/10.1016/j.anaerobe.2014.01.002)
Line 482-485 and line 496-501: these sentences focus on the same point which is the low probability of BoNT phage transfer based on the comparison phylogeny obtained with the whole genome analysis and the BoNT phage analysis. This should be gathered to avoid repetitions.
BoNT-gene-carrying plasmid clustering
Clustering obtained based on bont plasmid is very similar to the one obtained based on the chromosom, except for genome 49511 of group IA and 4 IZSVeTV 10380 (_13/14, _20/14, _18/14 and _17/14) of group IB. How should this be interpreted? Does it mean a recent acquisition of the bont phage? What are the hypothesis here to explain these exceptions? A paragraph about this should be added in the discussion.
As the new FliC genes were detected in 10380, is there any link between both results (FliC-VI and VII and difference in clustering)?
Strain 49511 from France has been classified as a singleton among lineage Ia, is their any link with the fact that clustering is different when using the chromosome or the BoNT phage?
BoNT gene sequence comparison
Line 506-510: have the sequence differences in the bont gene sequence any consequences on the functionality of the toxin? What is the impact of the amino acid substitution from glycine to valine in strain IZSVeTV_5262/1a/14 for example? The point here in the study was to show the high conservation of bont genes in group III C. botulinum on the contrary to group I or II but the impact on the functionality of toxin should also be addressed.
Plasmids
It seems that in strains isolated from the same outbreaks the plasmids are not always the same (Table S3): (for example, this is not exhaustive)
- IZSVeTV 10380 (_13/14, _20/14, _18/14 and _17/14) were isolated from outbreak ID1 but PG16 was detected only in 13/14 and 20/14 and not in 18/14 and 17/14.
- IZSVeTV1585/19/11 and 18/11, PG16 was found in 18/11 but not in 19/11. Moreover in 1671 (same outbreak) neither PG16 nor PG14 were detected. The dendrogram in Table S2 show however that there is not difference based on SNP analysis for the 3 strains suggesting that these strains are clonal.
- IZSVeTV 2454/6/14, IZSVeTV 2454/7/14, IZSVeTV 2667/2/14 from outbreak 3, PG14, PG20, PG22 and PG16 were found in some but not all the 3 genomes. Here again the dendrogram in Table S2 shows however that there is not difference based on SNP analysis for the 3 strains suggesting that these strains are clonal.
- For IZSVeTV_2659_1_10, IZSVeTV_2659_2_10, IZSVeTV_4879_20_10 (outbreak 4), PG21 is missing in IZSVeTV_4879_20_10
- For IZSVeTV_3522_4_13, IZSVeTV_3522_5_13, IZSVeTV_3524_3_13 (outbreak 5), PG16, PG20 and PG22 are not present in the 3 strains…
How do authors explain this? Are some plasmids unstable like the bont phage? Could this be explained by something that happened during DNA extraction or sequencing? Should a specific DNA extraction method dedicated to plasmid extraction be used for the study of C. novyi sensu lato plasmids ? Could please authors add something about this in the discussion?
Could please authors add the outbreak ID in Table S3?
Line 377-378: some plasmids are specifically detected in some clusters. This is not addressed in the discussion. Does this have any significance as far as evolution regards? Could it provide an advantage to strains that harbor these plasmids?
Minor comment:
Could authors please revise the article and delete the unnecessary “-“ like for example line 116 “Diag-nostic” or line 118 “to-gether”? This appears all along the text.
Could authors please put in italic the bacterial species?
Line 185: please change “diffence” by “difference”
Line 260: could “group IA” be replaced by “lineage IA” (it can be confused with C. botulinum group I)
Lines 87-92, 491, 492, 493, 495, 577, 578, line 87-92, lines 252-260, Line 266: same comment please replace “group” by lineage (when mentioning lineage Ia, Ib, II, III and IV).
Line 497: please delete the “t” in “transfert”
Line 503: please change “recombinat » by “recombinant” or “mosaic”
Line 540: please replace “take” by “taken”
Line 562: please replace “novy » by “novyi”
Author Response
SECOND REVIEWER
Comments and Suggestions for Authors
This study presents the results of sequencing of 60 C. botulinum group III isolated in Italy from 32 animal botulism outbreaks between 2008 and 2016. Before this study, only 58 genomes of C. novyi sensu lato were available in public databases (48 included in this study), demonstrating the great relevance and interest of this study by doubling the number of available genomes. The analysis of these genomes allowed the confirmation of the existence of 4 lineages in the C. novyi sensu lato group, of a large plasmidome and the relevance of the use of FliC genes for subtyping. New plasmids are described and two new FliC genes have been sequenced and fully compared to other previously described FliC genes. Moreover this study provides very useful information about epidemiology of animal botulism through source identification (Rice hulls for example), or by highlighting the risks of cross-contamination between wild birds and poultry as closely related strains are isolated in both cases.
The article is well written (minor typing errors) and easy to read.
Major comments:
Materials and methods:
Forty-eight genomes were included in this study. According to Le Gratiet et al. 2021, 58 genomes of C. novyi sensu lato are currently available. Moreover complete genomes of 4 strains were published in 2021 (Woudstra et al. 2021). Could authors please indicate when the genomes were retrieved here for their study so as to justify why the 10 other available genomes were not included in their study (line 142-143)? Citation of these two recent articles is not necessary (at least here) but to my point of view it is important to explain why all genomes that are currently available in public databases were not included.
R: the genomes included in the study were those available at the time in which the reported analyses were performed, that was prior to 2020.
Line 152-153: could authors please explain how they determine that culture was at the exponential growth phase (enumeration? Optical density? Other way?). How long was the incubation on average?
R: The sentence has been changed as follows: “Genomic DNA was extracted from a 48 h broth culture using …” (now line 159-160
The incubation time was 48 h, as reported in line 148.
Line 239-242: Most of the strains were isolated from the North-East of Italy. Is it because most animal outbreaks occurred in the region in Italy? Could please authors explain their choice about the distribution among the different regions? Why did they mostly sequence strains from this region and not select the same number of strains per region so as to have an equal distribution of the strains across the country? (This may be part of the discussion or in the MM section (2.1 strains) and not included in the results section)
R: Lines 239-242 have been deleted in section 3.1.
Otherwise, the Material and Methods (section 2.1) lines 127-132 have been modified as follow (now lines 127-135):
“The majority of the strains (52/60) were isolated in outbreaks occurred in regions of the North-East of Italy (Veneto, Friuli Venezia Giulia, Trentino Alto Adige and Emilia Romagna), three in regions of the North West (Lombardia), four in the Center and one in the South of Italy (Puglia) (Table 1). Most of the strains were isolated from the North-East of Italy because this is the area of territorial competence of the Istituto Zooprofilattico Sperimentale delle Venezie (IZSVE).”. Lines 239-242 have been deleted in section 3.1 .
Figure 1: could authors please add the reference of each outbreak in the dendrogram? The numbering is confusing, For example, the strain 3859/5/11 is part of outbreak 19 and not from outbreak 11. Addition of the outbreak number in the dendrogram will be very helpful for the reader.
R: The dendrogram is modified as requested. A column has been added with the outbreak ID.
Is it also possible to improve the quality of the Figure 1 and Figure 3 (they are blurry when zoomed in and numbers are difficult to read)?
R: the quality of the Figure 1 and Figure 3 has been improved.
Could authors please explain how they determined the subclusters within each lineage? Which criteria were used to distribute the strains in the different subclusters?
R: The subclusters were determined observing the branching of the dendrogram. In particular the main branches of hierarchy immediatey inferior to the cluster branches.
Table 2: is Table 2 really necessary in the article? Couldn’t it be considered as supplemental data?
R: Table 2 has been moved in supplementary data (Suppl. Table 1, sheet 1).
Flagellin: could it be possible to add the FliC group of each strain in one Table or Figure? (Table 1 for example? Or in Figure 1?)
R: In Table 2, a column has been added reporting the FliC type
SNPs analysis:
Table S2: How should the data in these tables be interpreted? Is this the number of SNPs that differs between two genomes or the % of difference between two strains? This is not detailed neither in the legend in the article nor above the tables.
R: the values in the tables represent the number of SNPs that differs between two genomes. The legend was rewritten to clarify the issue: “Table S2: subgroups of very similar genomes: SNP similarity matrices as determined by data obtained with the program kSNP [54].The values in the tables represent the number of SNPs that differs between two genomes. As indicated, for some subgroups two matrices are shown:...”
Line 286-288 and line 294 and line 298: Could authors please explain why they say that 300 SNPs is low and above 1000 is distant? What are the criteria to determine when the number of SNP is low or high?
R: the sentence in line 286-288 was rewritten in neutral forms: “From the analysis, it resulted that the genomes in these subgroups, isolated in European countries also very distant between them, differ for a number of SNPs < 300”. For the other sentences, see the answer to the following comment
Line 290-298: Regarding these results, some discussion is missing. Authors report between 1 and 3 SNPs for genomes of outbreaks 5, 11, 16 and 17. How should the presence of these SNPs be interpreted? Could these genomes be considered to be the same? Where are the SNPs located?
R: line 290-298 were rewritten as follows: “For 17 epidemiological events, more than one isolated was sequenced (Table 1). Except for outbreaks with ID7 and ID9, the strains isolated from the same outbreak resulted clones: 3 SNPs were identified in ID17, two SNPs in ID15, one in ID5, 11 and 16, while no genetic differences were detected in the other events (Table S2). Differently, in the events ID7 and ID9, the genetic distance (>1000 SNPs) makes the isolates in the same outbreak be considered different strains, though disposed in the same subcluster (subcluster 2 of lineage II and sub- cluster 2 of lineage IA, respectively) on Gegenees dendrogram (Figure 2).”
The authors believe that sequences that differ by a few SNPs (three o fewer) are likelythe same. It is possible that so few mutations arise during the time of an outbreak. All SNPs found in these cases are located in the chromosome, lying in coding sequences, except one located in an intergenic region of the BoNT-gene-carrying plasmid.
Line 448-449: this has also been mentioned in Anza et al. 2014 where strains from Spain and From Sweden were compared (http://dx.doi.org/10.1016/j.anaerobe.2014.01.002)
R: the reference has been added.
Line 482-485 and line 496-501: these sentences focus on the same point which is the low probability of BoNT phage transfer based on the comparison phylogeny obtained with the whole genome analysis and the BoNT phage analysis. This should be gathered to avoid repetitions.
R: the content of the sentences has been gathered: “Differently from the small plasmids, the big-sized plasmids, in particular the BoNT-gene-carrying plasmid, have shown to be unfrequently transferred by horizontal heredity mechanisms, since the phylogeny built on their sequences is the same of the whole genome sequence deduced phylogeny [16,42]. This observation is substantially confirmed by the present analysis, as it can be seen comparing the whole genome dendrogram (Figure 1) and BoNT-gene-carrying plasmid dendrogram (Figure 3); even if slight differences in topology between the two dendrograms do not allow to exclude that BoNT-gene- carrying plasmid lateral exchange is possible.” The new sentence was placed in the position of line 496-501.
BoNT-gene-carrying plasmid clustering
Clustering obtained based on bont plasmid is very similar to the one obtained based on the chromosom, except for genome 49511 of group IA and 4 IZSVeTV 10380 (_13/14, _20/14, _18/14 and _17/14) of group IB. How should this be interpreted? Does it mean a recent acquisition of the bont phage? What are the hypothesis here to explain these exceptions? A paragraph about this should be added in the discussion.
R: the authors have proposed a possible explanation in the discussion (line 518-520): ”...but slight differences regarding the placement of some strains in sub-clusters do not allow to exclude that BoNT-gene-carrying plasmid lateral exchange is possible”.
As the new FliC genes were detected in 10380, is there any link between both results (FliC-VI and VII and difference in clustering)?
R: the two facts could be not casually linked. But the logic of the link is not evident, and the elaboration of speculations explaining the link would require further analysis (for example, the search of signs of homolog recombination on the chromosome of 10380 genomes) that are not among our priorities at present.
Strain 49511 from France has been classified as a singleton among lineage Ia, is their any link with the fact that clustering is different when using the chromosome or the BoNT phage?
R: this could be the signature of homologous gene transfer events involving the BoNT phage, as told in the Discussion.
BoNT gene sequence comparison
Line 506-510: have the sequence differences in the bont gene sequence any consequences on the functionality of the toxin? What is the impact of the amino acid substitution from glycine to valine in strain IZSVeTV_5262/1a/14 for example? The point here in the study was to show the high conservation of bont genes in group III C. botulinum on the contrary to group I or II but the impact on the functionality of toxin should also be addressed.
R: As reported in Materials and Methods, all the strains underwent to the mouse bioassay to confirm that their toxicity was conserved after the defrosting. Strain IZSVeTV_5262/1a/14 showed to be toxic for mice, but we did not investigate the amount of BoNTs produced.
The authors add lines 347-348 in Results,: ”No reduction in toxicity was observed during the mouse bioassay for the strains with one aa substitution”.
It seems that in strains isolated from the same outbreaks the plasmids are not always the same (Table S3): (for example, this is not exhaustive)
- IZSVeTV 10380 (_13/14, _20/14, _18/14 and _17/14) were isolated from outbreak ID1 but PG16 was detected only in 13/14 and 20/14 and not in 18/14 and 17/14.
- IZSVeTV1585/19/11 and 18/11, PG16 was found in 18/11 but not in 19/11. Moreover in 1671 (same outbreak) neither PG16 nor PG14 were detected. The dendrogram in Table S2 show however that there is not difference based on SNP analysis for the 3 strains suggesting that these strains are clonal.
- IZSVeTV 2454/6/14, IZSVeTV 2454/7/14, IZSVeTV 2667/2/14 from outbreak 3, PG14, PG20, PG22 and PG16 were found in some but not all the 3 genomes. Here again the dendrogram in Table S2 shows however that there is not difference based on SNP analysis for the 3 strains suggesting that these strains are clonal.
- For IZSVeTV_2659_1_10, IZSVeTV_2659_2_10, IZSVeTV_4879_20_10 (outbreak 4), PG21 is missing in IZSVeTV_4879_20_10
- For IZSVeTV_3522_4_13, IZSVeTV_3522_5_13, IZSVeTV_3524_3_13 (outbreak 5), PG16, PG20 and PG22 are not present in the 3 strains…
How do authors explain this? Are some plasmids unstable like the bont phage? Could this be explained by something that happened during DNA extraction or sequencing? Should a specific DNA extraction method dedicated to plasmid extraction be used for the study of C. novyi sensu lato plasmids ? Could please authors add something about this in the discussion?
R: at line 379, the following comment has been added: “in some case, strains isolated in the same outbreak, that resulted equal by SNP analysis, do not appear to contain the same plasmids (supplementary table 2, sheet 1). This can be due to low stability of the plasmid through cell division, but could also be the mere effect of plasmid loss during DNA extraction or gaps in sequencing”.
Could please authors add the outbreak ID in Table S3?
R: Outbreak ID has been added in the table S3
Line 377-378: some plasmids are specifically detected in some clusters. This is not addressed in the discussion. Does this have any significance as far as evolution regards? Could it provide an advantage to strains that harbor these plasmids?
R: for an exhaustive answer to the posed questions, more details are needed on fenotipic, pathogenic and ecological features of the various clusters, that at this time we have not examined. Such questions could be the object of a further study.
Minor comment:
Could authors please revise the article and delete the unnecessary “-“ like for example line 116 “Diag-nostic” or line 118 “to-gether”? This appears all along the text.
R: the unnecessary hyphenation has been removed
Could authors please put in italic the bacterial species?
R: Bacterial species are now in italic
Line 185: please change “diffence” by “difference”
R: It was modified
Line 260: could “group IA” be replaced by “lineage IA” (it can be confused with C. botulinum group I)
R: It was modified
Lines 87-92, 491, 492, 493, 495, 577, 578, line 87-92, lines 252-260, Line 266: same comment please replace “group” by lineage (when mentioning lineage Ia, Ib, II, III and IV).
R: It was modified
Line 497: please delete the “t” in “transfert”
R: done
Line 503: please change “recombinat» by “recombinant” or “mosaic”
R: It was modified
Line 540: please replace “take” by “taken”
R: It was modified
Line 562: please replace “novy » by “novyi”
R: It was modified